# Different Cryotherapy Modalities Demonstrate Similar Effects on Muscle Performance, Soreness, and Damage in Healthy Individuals and Athletes: A Systematic Review with Metanalysis

**DOI:** 10.3390/jcm11154441

**Published:** 2022-07-30

**Authors:** Klaus Porto Azevedo, Júlia Aguillar Ivo Bastos, Ivo Vieira de Sousa Neto, Carlos Marcelo Pastre, Joao Luiz Quagliotti Durigan

**Affiliations:** 1Rehabilitation Sciences Program, Physical Therapy Division, University of Brasilia, Brasília 72220-275, Brazil; klauspa13@gmail.com (K.P.A.); juliaaguillar.ib@gmail.com (J.A.I.B.); 2Health Technologies Science Program, University of Brasília, Brasília 72220-275, Brazil; ivoneto04@hotmail.com; 3Physical Therapy Department, Paulista State University, Presidente Prudente 19060-900, Brazil; marcelo.pastre@unesp.br

**Keywords:** physical agents, physical therapy, rehabilitation, recovery

## Abstract

Background: There are extensive studies focusing on non-invasive modalities to recover physiological systems after exercise-induced muscle damage (EIMD). Whole-body cryotherapy (WBC) and Partial-body cryotherapy (PBC) have been recommended for recovery after EIMD. However, to date, no systematic reviews have been performed to compare their effects on muscle performance and muscle recovery markers. Methods: This systematic review with metanalysis compared the effects of WBC and PBC on muscle performance, muscle soreness (DOMS), and markers of muscular damage following EIMD. We used Pubmed, Embase, PEDro, and Cochrane Central Register of Controlled Trials as data sources. Two independent reviewers verified the methodological quality of the studies. The studies were selected if they used WBC and PBC modalities as treatment and included muscle performance and muscle soreness (DOMS) as the primary outcomes. Secondary outcomes were creatine kinase and heart rate variability. Results: Six studies with a pooled sample of 120 patients were included. The methodological quality of the studies was moderate, with an average of 4.3 on a 0–10 scale (PEDro). Results: Both cryotherapy modalities induce similar effects without difference between them. Conclusion: WBC and PBC modalities have similar global responses on muscle performance, soreness, and markers of muscle damage.

## 1. Introduction

Studies have extensively focused on non-invasive and viable strategies for optimizing the recovery of physiological systems and functional reestablishment associated with muscle damage [1,2,3]. Cryotherapy is a physical therapy strategy applied for acute musculoskeletal system injuries and athletic performance recovery [4]. Many modalities of cryotherapy application have been recommended for the recovery of physiological systems after exercise-induced muscle damage (EIMD), including whole-body cryotherapy (WBC). This is a cryotherapy modality that involves the whole body and not necessarily just the neck and head; it can involve immersion or cold chamber applications. Partial-body cryotherapy (PBC) is a cryotherapy modality that involves only lower limbs, or an immersion until the iliac spine, ice pack application, and cold chamber [1,3,5]. A prior systematic review with meta-analysis showed that different cryotherapy strategies could be superior to passive recovery after various exhaustive exercise protocols for improving DOMS [6]. However, the authors did not find the best modality for muscle performance (Vertical Jump Performance (VJP) and Maximal Voluntary Isometric Contraction (MVIC)), muscle soreness (Delayed Onset Muscle Soreness (DOMS)), and biochemical markers of muscular damage (Creatine Kinase (CK)) [6].

The rationale for cryotherapy application after muscle damage has developed over the years due to various positive physiological effects, including inflammatory, vascular, neurological, and metabolic adaptations [4,7,8,9]. The magnitude of these mechanisms depends on the intensity and time of the cold application and how it impacts the body. Machado et al. [7] showed findings that DOMS has the best improvement with cryotherapy application with temperature between 11 and 15 °C when applied for 11 to 15 min. Furthermore, protocols that involve the body and limbs [10,11,12] (i.e., WBC) may have an advantage in increasing parasympathetic reactivation post-exercise compared to other applications. WBC protocols may have a positive influence on skin temperature uniformity and hydrostatic pressure, causing vasoconstriction and higher cellular activation in multiple physiological systems [2,5]. In contrast, the mechanism underpinning WBC application with cold chambers or cryocabins did not consider hydrostatic pressure [13], because the application consists of exposure to very cold air [14].

Investigations have suggested that cryotherapy improves delayed onset muscle soreness (DOMS) regardless of modality [15,16]. Studies also indicate that both cryotherapy strategies (WBC or PBC) improved muscle performance in different exercise types and alleviated muscle damage [5,6]. However, controversy regarding the effectiveness of each modality is still presented. The differences between study results could be explained by the heterogeneity of muscle damage protocols and exercise modalities [1,12,17,18,19]. In addition, heart rate variability (HRV) is a non-invasive tool used to obtain valuable data concerning physiological changes in response to exercise [20]. Changes in the patterns of HRV may serve as helpful parameters for post-exercise recovery, and this may complement more invasive tests. Interestingly, a recent study by Chauvineau et al. [20] showed that WBC increased nocturnal parasympathetic modulation compared with PBC and control conditions, which may result in overall recovery optimization and a healthier status [20].

According to these conflicting results, knowing the impact of different cryotherapy applications on the neuromuscular system would contribute to evidence-based practice and the effective design of recovery interventions. Therefore, this systematic and metanalysis review aimed to examine and compare the effects of WBC and PBC on muscle performance, damage, and soreness, as well as on HRV. Considering the heterogeneity in the literature, we hypothesized that WBC and PBC modalities would produce similar effects on muscle performance and markers of muscle damage. Our findings may help physical therapists better understand the impact of different clinical-like cryotherapy modalities on muscle damage and design more rational stimulation strategies.

## 2. Materials and Methods

### 2.1. Preliminary Settings

This systematic review was prospectively registered in the International prospective register of systematic reviews—PROSPERO (registration number CRD42021246443). The set of items of this systematic review is presented according to the Preferred Reporting Items for Systematic Reviews and Meta-Analysis statement (PRISMA) (Appendix A). The study question and other systematic review procedures addressed the following PICOT: population: healthy individuals and athletes; intervention: whole-body cryotherapy; comparisons: partial body cryotherapy; outcomes: DOMS; CK; MVIC; VJP; and HRV. Study protocols commonly applied three different follow-ups to assess the outcomes: immediately-post intervention, 24 h after the intervention, and 48 h after the intervention.

### 2.2. Eligibility Criteria

We included randomized controlled trials (RCTs) that compared whole-body and partial-body cryotherapy in healthy individuals. The primary outcomes assessed were DOMS, VJP, and MVIC. The secondary outcomes were CK and HRV. Studies were excluded from the analysis based on the following criteria: (i) clinical trial registers or non-concluded studies, dissertations and theses letters to the editor, reviews, and observational studies; (ii) clinical trials not related to the object of study; (iii) inclusion of passive comparators such as placebo and or sham therapy or active comparator such as another intervention; (iv) clinical trials with individuals with neurological or musculoskeletal impairment; (v) clinical trials with subjects who participated in any regular training or physical activity program during cryotherapy; (vi) unable to find full version; (vii) missing data. Two independent reviewers selected the studies. Any disagreement between the reviewers was resolved by consensus, and if necessary, a third reviewer was asked to decide on the inclusion of the studies.

### 2.3. Search Strategy

The study selection process included the screening of titles, reading of abstracts, checking for duplicated studies, evaluating inclusion criteria, and full-text reading. A literature search was conducted from November 2020 to August 2021 in the following databases: Pubmed, Embase, Web of Science, EBSCO, PEDro, Cochrane Central Register of Controlled Trials (CENTRAL), and SCOPUS. The search terms used were based on the strategies suggested by the Cochrane Handbook for Systematic Reviews of Interventions, and were: (“athletes”[MeSH Terms] OR “healthy volunteers”[MeSH Terms] OR “healthy participants”[Title/Abstract]) AND (“cryotherapy”[MeSH Terms] OR “whole body cryotherapy”[Title/Abstract] OR (“immersion/therapeutic use”[MeSH Terms] OR “partial body cryotherapy”[Title/Abstract] OR “water immersion”[Title/Abstract] OR “cold temperature/therapeutic use”[MeSH Terms] OR “cold temperature/therapy”[MeSH Terms] OR “cold pack”[Title/Abstract] OR “ice pack”[Title/Abstract] OR “partial body cryotherapy”[Title/Abstract] OR “cold water immersion therapy”[Title/Abstract] OR “cold water immersion treatment”[Title/Abstract] OR “cold water therapy”[Title/Abstract] OR eccentric[Title/Abstract] OR cold air[Title/Abstract] OR hydrotherapy[Title/Abstract] OR contrast water therapy[Title/Abstract] OR Water immersion[Title/Abstract]) AND (“delayed onset muscle soreness”[Title/Abstract] OR “pain”[MeSH Terms] OR “soreness”[Title/Abstract] OR “creatine kinase”[MeSH Terms] OR “isokinetic”[Title/Abstract] OR control, heart rate[MeSH Terms] OR heart rate control[MeSH Terms] OR “torque”[MeSH Terms] OR “peak torque”[Title/Abstract] OR “strength”[Title/Abstract] OR “jump performance”[Title/Abstract] OR “countermovement jump” OR “autonomic response”[Title/Abstract]). To find a greater number of articles, these terms were adjusted in each database. There were no restrictions regarding the language and date of publication of the potentially eligible studies.

### 2.4. Data Extraction and Analysis

Data related to the number and characteristics of participants (sample size, subject’s age, and gender) were extracted; and year of publication; description of the intervention (application characteristics (cold chambers, cryocabins, cold water immersion) treatment area, and duration); primary outcomes; and tools used to evaluate results and study results. Data to perform the meta-analysis were extracted by one reviewer and checked simultaneously by a second reviewer. Values were entered into a database on Excel Software before using Review Manager software (Version 5.4.1; Cochrane library; United Kingdom; September 2020; available in: https://training.cochrane.org/online-learning/core-software/revman/non-cochrane-reviews).

The standardized Mean Difference (SMD: measures the absolute difference between the mean values in two groups in a clinical trial) and 95% Confidence Intervals were considered. For this analysis, we used the mean difference between baseline and the measurements immediately-post, 24 h after, and 48 h after the intervention in the metanalysis. The authors of the included studies were contacted to request any necessary additional information on the data.

The included papers had distinct populations, intervention parameters, and settings, so a random-effects model (inter-study heterogeneity) was always employed in the meta-analysis. We determined the statistical heterogeneity of data with an I2 test and interpreted the results considering values above 25 and 50% as moderate and high heterogeneity, respectively **[21]**. A *p*-value < 0.05 was considered significant. All analyses were conducted using Review Manager Software version 5.4.1

### 2.5. Quality Assessment of the Studies

To evaluate the quality of the included studies, two authors independently assessed the selected studies using two instruments. The 11-item PEDro scale, which quantitatively includes the following 11 items: (1) eligibility criteria were specified (not used to calculate score); (2) subjects were randomly allocated to groups; (3) allocation was concealed; (4) the groups were similar at baseline regarding the most important prognostic indicators; (5) there was blinding of all subjects; (6) there was blinding of all therapists who administered the therapy; (7) there was blinding of all assessors who measured at least one key outcome; (8) measures of at least one key outcome were obtained from more than 85% of the subjects initially allocated to groups; (9) all subjects for whom outcome measures were available received the treatment or control condition as allocated or, where this was not the case, data for at least one key outcome was analyzed by “intention to treat”; (10) the results of between-group statistical comparisons are reported for at least one key outcome; and (11) the study provides both point measures and measures of variability for at least one key outcome. Each of the items was marked as “yes (1)” or “no (0)” and the final score was on a scale from 0 to 10. Studies were considered high quality if they received scores equal to or greater than 6. Studies with scores lower than 6 were deemed to be low quality [22].

### 2.6. Quality of Evidence

The overall quality of the evidence was rated in accordance with the Grading of Recommendations, Assessment, Development, and Evaluation (GRADE) **[23]**. The GRADE has five domains to establish the quality of evidence: (1) Study design and risk of bias; (2) Inconsistency; (3) Indirectness; (4) Imprecision; and (5) Other factors (e.g., reporting bias, publication bias). The quality of the evidence was classified as follows: High-quality evidence: when there were consistent results in at least 75% of the clinical trials of good methodological quality, presenting consistent, direct, and precise data with no suspicion of or known publication bias. Further research is unlikely to alter the estimate or the confidence in the results; Moderate quality of evidence: When at least one domain is not met. New research is likely to significantly alter the confidence in the effect estimate; Low-quality evidence: When two of the domains are not met. Further research is likely to significantly impact the confidence in the effect estimate and is likely to alter the estimate; Very low-quality evidence: When three domains are not met, the results will be highly uncertain **[24]**.

## 3. Results

A total of 120 subjects were evaluated from 2013 to 2021. The search generated a total of 4414 potentially eligible articles. Seven trials were considered eligible after applying the inclusion criteria and were included in the review (Figure 1).

Of these six included studies, two of them analyzed athletes (*n* = 42), two of them analyzed physically active subjects (*n* = 30), and two studies analyzed well-trained individuals (*n* = 48). All of the characteristics of the studies are presented in Table 1.

Five trials assessed the MVIC, all of them through the mean force. Two studies assessed jump performance through the vertical jump test. Five studies evaluated the DOMS outcome using a ten-centimeter analogue scale. Three studies assessed CK levels through blood samples. Only one study assessed heart rate variability; this study was included for qualitative analysis. Of the six included studies, three of them [12,13,25] analyzed WBC as a cold chamber application.

### 3.1. Methodological Quality Assessment

The methodological quality of the included studies using the PEDro score is presented in Table 2. The mean PEDro score was 4.3 for all the assessed papers, on a scale from 0 to 10.

### 3.2. Primary Outcomes

#### Maximal Voluntary Isometric Contraction (MVIC)

The five trials that analyzed the force used the peak torque as an outcome, represented as the peak of evoked torque on MVIC of knee extensors, with very low-quality evidence (downgrade by the risk of bias, imprecision, and inconsistency). WBC and PBC modalities have similar effects on MVIC (Figure 2A–C). MVIC data were analyzed in three different follow-up moments, immediately post-intervention and 24 and 48 h after the intervention. The meta-analysis results are shown in Figure 2A for immediately post with SMD: 0.03 and 95% CI: −€€€€0.59, 0.65, *p* = 0.94, Figure 2B for 24 h with MD: −0.13 and 95% CI −0.53, 0.27, *p* = 0.53, and Figure 2C for 48 h with MD: −0.82 and 95% CI −9.47, 7.83 *p* = 0.85.

### 3.3. Vertical Jump Performance (VJP)

According to GRADE, the three studies that analyzed jump performance used the vertical jump test as an outcome, represented by the height of the jump, with very low quality (downgraded by inconsistency and imprecision). WBC and PBC modalities have similar effects on VJP (Figure 3A–C). VJP data were analyzed in three different follow-ups, immediately post-intervention and 24 and 48 h after the intervention. The meta-analysis results are shown in Figure 3A for immediately post with SMD: −0.20 and CI 95%: −0.85, 0.45, *p* = 0.54, Figure 3B for 24 h with SMD: −0.11 and CI 95%: −0.73, 0.51, *p* = 0.54, and Figure 3C for 48 h with SMD: −0.17 and CI 95%: −0.79, 0.45, *p* = 0.59.

### 3.4. Delayed Onset Muscle Soreness (DOMS)

Five studies assessed DOMS using the Visual Analogue Scale (VAS) as an index, with all samples together and low-quality evidence (downgraded by inconsistency and imprecision). WBC and PBC modalities have similar effects on DOMS (Figure 4A,B). DOMS data were analyzed in two different follow-ups, 24 and 48 h after the intervention. The meta-analysis results are shown in Figure 4A for 24 h with SMD: −0.25 and CI 95%: −0.66, 0.15 *p* = 0.23, and Figure 4B for 48 h with SMD: −0.08 CI 95%: −0.48 to 0.33, *p*: 0.72.

### 3.5. Secondary Outcomes

#### CK Levels

The three papers that analyzed creatine kinase levels used blood levels of CK as an outcome, with very low-quality evidence (downgraded for inconsistency, indirectness, and imprecision). WBC and PBC modalities have similar effects on CK (Figure 5A–C). CK data were analyzed in three different follow-ups, immediately post-intervention and 24 and 48 h after the intervention. The meta-analysis results are shown in Figure 5A for immediately post with SMD: −0.22 and CI 95%: −1.09, 0.66, *p* = 0.72, Figure 5B for 24 h with SMD: 0.26 and CI 95%: −0.27, 0.80, *p* = 0.33, and Figure 5C for 48 h with SMD: 0.05 and CI 95%: −0.47, 0.58, *p* = 0.84.

### 3.6. Heart Rate Variability (HRV)

None of the included papers evaluated the heart rate variability (HRV) outcome.

## 4. Discussion

This meta-analysis review summarizes the current evidence on WBC and PBC modalities used primarily for perceived soreness, athletic performance, and muscle damage in healthy individuals and athletes. Both cryotherapy modalities induced similar effects on muscle performance, soreness and damage, without difference between them, allowing physical therapists and physiologists to choose cryotherapy modalities according to their practice context. Considering these findings, we are not able to conclude that the hydrostatic pressure physiological mechanism considered in the WBC application can induce significant differences in VJP, MVIC, DOMS, and CK levels. Therefore, physiotherapists could choose WBC or PBC and expect similar treatment effects. Physiotherapists can choose the modality that could be more feasible according to their available resources and application possibilities. Nonetheless, according to the GRADE recommendation [22], the quality of evidence was very low for muscle performance, DOMS, and damage. Thus, new findings may alter the conclusions presented in this review, and the present results should be interpreted cautiously. The findings presented here have practical implications for sports medicine and rehabilitation purposes.

In recent years, the number of studies focusing on cryotherapy has increased, and significant systematic reviews have been performed to compare the effects of WBC and other soreness recovery strategies [5,12,13,18,20,25]. However, these systematic reviews [6,7,26,27] aimed to explore the beneficial effects of cryotherapy on muscle soreness, regardless of their application modality. The muscle soreness reduction after WBC and PBC could be justified by the factors that mediate the analgesia process, which lead to pain–spasm cycle reduction [28]. Cold application, in the case of immersion, seems to modify hydrostatic pressure and decrease the ability of sensory transmission and thus reduce acetylcholine release, influencing the pain threshold [29]. In the current study, we observed similar effects on perceived soreness in both WBC and PBC strategies, suggesting that the approach chosen should consider patient compliance and logistics aligned with adhesion to long-term applications. One possibility is that PBC will probably be more feasible considering that patients do not need to be immersed up to the shoulder or neck [30].

Two out of six studies demonstrated similar results in VJP regardless of the modality type (WBC or PBC). Previous investigations show that VJP values decreased in the first hours post-exercise, which is explained by the fatigue resulting from the EIMD. However, 24 h after exercise, VJP values returned to baseline [5,16]. Furthermore, two investigations that compared WBC and PBC with the control group indicate that both interventions improved VJP values post-exercise [5,16]. Although the vertical impulse is a substantial jump height predictor and can be considered the standard evaluation of peak force and power [31], muscle strength development is underpinned by a combination of several morphological and neural factors. Of interest, general and specific sports skills and their underpinning strength characteristics should be considered to determine the cryotherapy strategy and critical time-point for inclusion in the recovery.

In the meta-analysis results, no difference was highlighted between WBC and PBC for MVIC. In some of the investigations analyzed in this review, it was concluded that WBC could be harmful to MVIC, considering that it could provide a greater decrease in nerve velocity conduction and, consequently, a longer time for MVIC to return to baseline [15,23]. When designing the cryotherapy strategy using MVIC, to improve reliability and validity, we must consider the subjects’ characteristics, tasks, and target muscles. MVIC may be affected by internal effects, such as muscle fatigue, synergist contribution, gender, and motivation, which may have contributed to the conflicting results presented here [32,33].

A recent study showed that CK levels decreased faster after WBC intervention [12]. In contrast, other researchers concluded that WBC had a harmful effect on CK levels compared to the PBC modality [13,25]. The plausible mechanism involving CK response in the blood circulation could be explained by the perception of cold as a noxious stimulus, increasing inflammatory response, metabolic activity, and the regeneration process linked to muscle damage [34]. These controversial studies are probably due to different exercise protocols, such as eccentric [13]; marathon [25]; intermittent, and simulated trials [15,20]. Considerable variability in CK response could also be related to training status, muscle quality (muscle strength and hypertrophy), exercise modality (resistance, aerobic, or combined), intensity (light, moderate, or intense), frequency (days per week), and duration (session time) [34]. Of note, investigation of other proteins and regulatory molecules that participate in muscle damage should be investigated in future studies.

It is important to emphasize that the studies included in this review did not have high methodological quality, with a mean PEDro score of 4.2 [22]. The evidence is considered of very low quality for VJP and MVIC. However, DOMS and CK display low quality. Further studies are needed with exercise protocol standardization to improve evidence quality concerning muscle damage induction. Moreover, the variability between WBC and PBC (duration of application, temperature, and athletic experience) and time-points for evaluations precluded us from performing meta-analyses for some outcomes, limiting this review to descriptive rather than quantitative comparisons. Finally, the database search yielded studies predominantly in English-language journals and may not have captured studies in non-English journals and regional databases.

It was not possible to perform a metanalysis for HRV analysis because no investigations were found comparing WBC and PBC pre and post-intervention. Nonetheless, it is essential to highlight that Chauvineau et al. [20] observed that WBC induced more significant thermal stress and nocturnal parasympathetic modulation than PBC and control conditions. These findings are relevant because cardiac parasympathetic activity during recovery from exercise may indicate the performance of a high-intensity exercise the following days [35,36]. A decrease in parasympathetic activity in response to WBC may be hypothetically harmful to elite athletes in the setting of consecutive high-performance activities. Different cooling strategies at several time-points post-exercise are required to identify the most effective approach.

Few clinical and physiological criteria have been established to define partial-body and whole-body cryotherapy on the analyzed outcomes. This gap is probably explained by the lack of biological plausibility involving dose-response. The optimal parameters of time of application, temperature, and body level are unclear. Given the complexity of biological systems, basic science may provide a meaningful understanding of the mechanisms involved when undergoing acute and chronic cryotherapy. Functional, morphological, and biochemical approaches might aid understanding of the full picture of muscle adaptation in response to WBC and PBC.

Our results open new perspectives for studies based on cryotherapy types, but due to the small number of studies included in the meta-analysis, sensitivity analyses and/or sub-group analyses were not possible. Another limitation to consider was that conclusions from the narrative synthesis had to be drawn from a small number of participants. Secondarily, the inclusion criteria of our research strategy included studies of cold chambers and whole-body immersion in the WBC group. This fact can be considered a limitation of this study since cold air is not an immersion protocol. Therefore, there is no effect of hydrostatic pressure [13], because the application consists of exposure to very cold air [14]. Nevertheless, these inclusion criteria agree with our search strategy prospectively registered at PROSPERO, considering the anatomic body segment receiving cold application as a whole-body modality. Thus, given the rapid development of research in this area, annual updates of this review are needed to keep pace with the latest findings regarding different cryotherapy types effectiveness and safety for athletes and healthy individuals.

## 5. Conclusions

Current evidence suggests that WBC and PBC have similar global effects on MVIC, VJP, DOMS, and CK levels in healthy individuals and athletes. Clinicians and physiologists could choose the more viable application considering their practice context. A dose-response relationship seems to be relevant considering DOMS and temperature and time of application should be monitored to induce better results regarding DOMS. Nevertheless, the overall methodological quality of the current literature is heterogenic in several key fields. Future larger, well-designed, and standardized investigations are needed to establish the optimal parameters to modulate muscle adaptations.

## Figures and Tables

**Figure 1 jcm-11-04441-f001:**
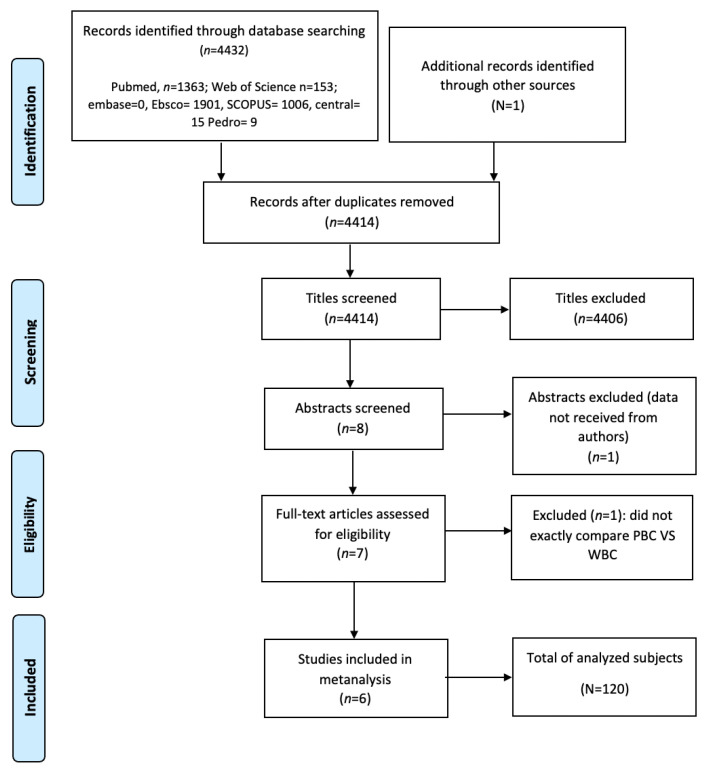
Prisma flowchart diagram. WBC: Whole-body cryotherapy (WBC); PBC: Partial-body cryotherapy.

**Figure 2 jcm-11-04441-f002:**
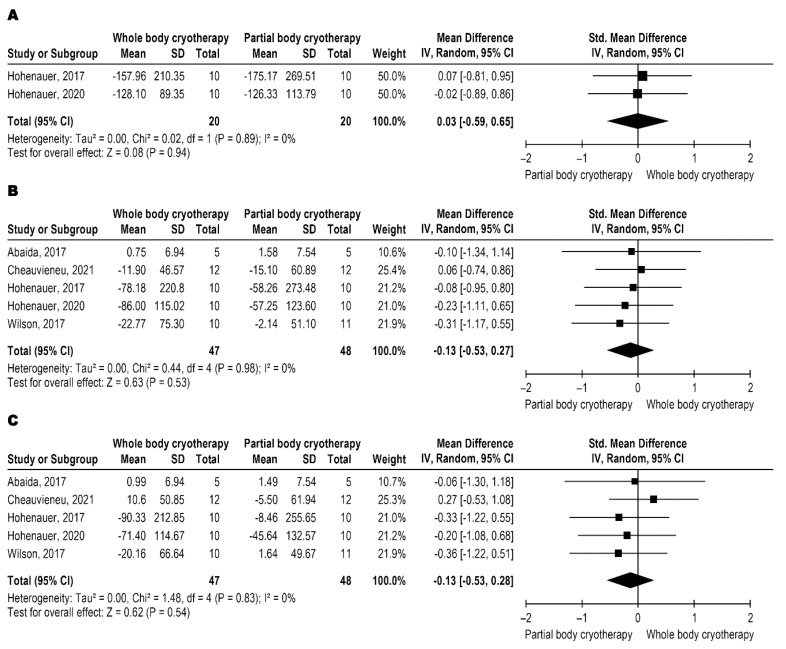
Metanalysis comparison between WBC and PBC for MVIC (**A**) immediately-post intervention [5,16], (**B**) 24 h after intervention [5,13,16,20,25], and (**C**) 48 h after intervention [5,13,16,20,25].

**Figure 3 jcm-11-04441-f003:**
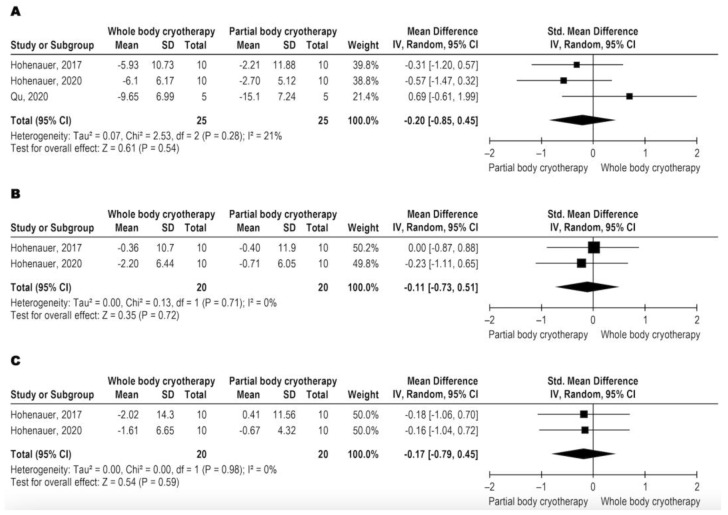
Metanalysis comparison between WBC and PBC for VJP (**A**) immediately-post intervention [5,12,16], (**B**) 24 h after intervention [5,16], and (**C**) 48 h after intervention [5,16].

**Figure 4 jcm-11-04441-f004:**
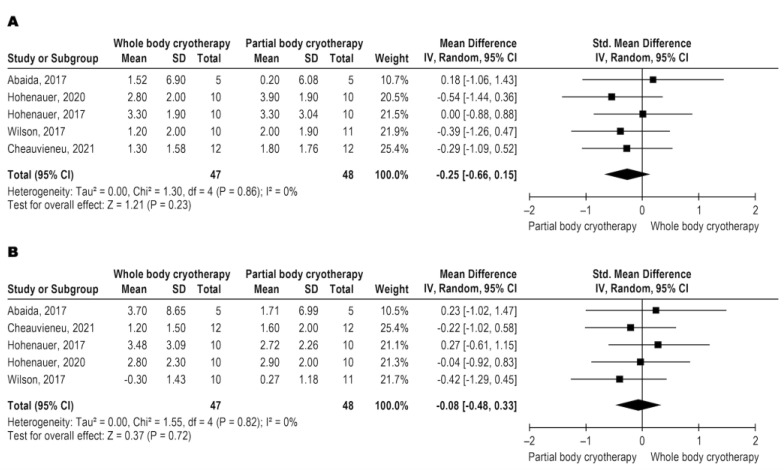
Metanalysis comparison between WBC and PBC for DOMS (**A**) 24 h after intervention and (**B**) 48 h after intervention [5,13,16,20,25].

**Figure 5 jcm-11-04441-f005:**
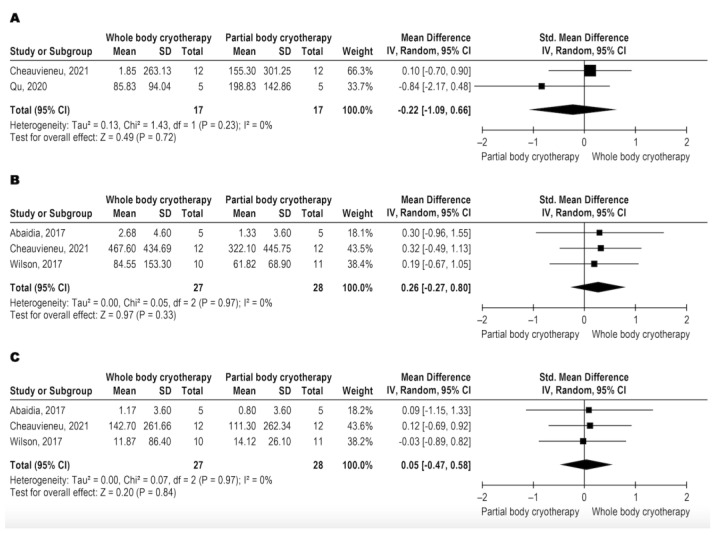
Metanalysis comparison between WBC and PBC for CK (**A**) immediately-post intervention [12,20], (**B**) 24 h after intervention [13,20,25], and (**C**) 48 h after intervention [13,20,25].

**Table 1 jcm-11-04441-t001:** Papers included in the quantitative analysis.

Authors	Study Design	Sample Size	Age (Years)	Whole-Body Protocol (Temperature and Time)	Partial Body Protocol (Temperature and Time)	Main Outcomes
Hohenauer et al. [5]	RCT ^1^	28	22.5	10 °C—10 min	30 s –60 °C; 2 min −135 °C	MVIC ^3^; VJP ^5^; DOMS^4^
Qu et al. [12]	RCT ^1^	20	21.0	−110 °C to −140 °C—3 min	15 °C—12 min	DOMS ^4^; CK ^2^; VJP ^5^
Abaidia et al. [13]	RCT ^1^	10	23.4	−110 °C—3 min	10 °C ± 0.3 °C—10 min	CK ^2^; MVIC ^3^; DOMS ^4^
Hohenauer et al. [16]	RCT ^1^	19	25.9	10 °C ± 0.37 °C—10	min −60 °C—30 s; −135 °C—2 min	MVIC ^3^; VJP ^5^; DOMS ^4^
Chauvineau et al. [20]	RCT ^1^	12	28.0	13.3 ± 0.2 °C—10 min	13.3 ± 0.2 °C—10 min	MVIC ^3^; CK ^2^
Wilson et al. [25]	RCT ^1^	31	39.8	−85 °C ± 5 °C—3 min	8 °C ± 0.5°—10 min	DOMS; CK ^2^; MVIC

Abbreviations: ^1^ Randomized Clinical Trial; ^2^ Creatine Kinase; ^3^ Maximal Voluntary Isometric Contraction; ^4^ Delayed onset Muscle Soreness; ^5^ Vertical Jump Performance.

**Table 2 jcm-11-04441-t002:** Methodological quality of the included articles (PEDro scale).

Author (Year)	Random Allocation	Concealed Allocation	Groups Similar at Baseline	Subject Blinding	Therapist Blinding	Assessor Blinding	Adequate Follow-Up	Intention-to-Treat Analysis	Between-Group Comparisons	Point Estimate and Variability	Total
Hohenauer et al. [5]	Y	N	Y	N	N	N	Y	N	Y	Y	5
Qu et al. [12]	Y	N	Y	N	N	N	Y	N	Y	Y	4
Abaidia et al. [13]	Y	N	Y	N	N	N	Y	N	Y	Y	4
Hohenauer et al. [16]	Y	N	Y	N	N	N	Y	N	Y	Y	5
Chauvineau et al. [20]	Y	N	Y	N	N	N	N	N	Y	Y	4
Wilson et al. [25]	Y	N	Y	N	N	N	N	N	Y	Y	4

Table 1: PEDro scale score of included studies in metanalysis.

## Data Availability

Not applicable.

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
