# Peer review of "Different Cryotherapy Modalities Demonstrate Similar Effects on Muscle Performance, Soreness, and Damage in Healthy Individuals and Athletes: A Systematic Review with Metanalysis"

_jcm, 2022, doi:10.3390/jcm11154441_

Round 1
Reviewer 1 Report
Manuscript Number: jcm-1725941
Title: Different cryotherapy modalities demonstrate similar effects on muscle performance, soreness, and damage in healthy individuals and athletes: a systematic review with metanalysis
Overview and general recommendation:
Initially, I would like to thank you for your submission and the opportunity to analyze the paper. This article addresses a relevant and well-known topic in the field of sports physiotherapy. Although the authors present a "small" systematic review with few studies and low-quality evidence, it is important to note that it is well written and presents the reader with a critical summary of the current evidence on the topic.
However, I believe that some points need to be better presented and addressed in the text. I explain my concerns in more detail below. I ask that the authors specifically address each of my comments in their responses.
Introduction:
The introduction is well written, presents information about what you know about the topic, gaps and inconsistency. However, the authors create an expectation about HRV, which throughout the paper is frustrated.
Do you consider it important to address this outcome in this review? During the planning of the review, did you not check if there were any more studies that had analyzed HRV?
Did the cited study (Chauvineau, 2021) not meet the eligibility criteria? This study is included in the review (Table 1).
Methods
I consider that the method is adequate and has been reported clearly and objectively.
Results
- The authors present the number of subjects evaluated. Then they present the information regarding the flowchart. Soon after, they present information about the subjects. I suggest presenting the flowchart data and then presenting the subjects' information.
- Correct the numbering of tables. You presented two tables 1.
- The databases presented in the flowchart differ from the databases presented in the method. Some bases are missing.
- Add information to table 1 (meaning of abbreviations).
- I believe that more information about the studies is needed. Improve the description of studies. For DOMS and MVC, which muscles were analyzed? Quadriceps? Triceps surae? Both? others?
- Although the authors have described in the method that they perform analysis with random effects, it is possible to observe in Figures 2A and 5A an analysis with fixed effects.
- Finally, the point that I consider most critical in the results session refers to the method used for meta-analysis. The authors present results using MD. However, I doubt whether all studies used the same scale to assess MVC. Some authors used the value in N, others in Nm, and others presented normalized values (%). In this case, it is correct to use the SMD method for meta-analyses. I suggest checking this point for all outcomes.
- Regarding HRV, the information presented is incorrect. Although it is not possible to carry out a meta-analysis, I suggest that the authors present a qualitative description of the findings of Maxime Chauvineau's study.
Discussion/ Conclusion
I believe that the discussion and conclusion are clearly presented. However, it is necessary to verify if with the correction of the analyzes it will be necessary to adjust some information.
Author Response
Reviewer 1
Overview and general recommendation:
Initially, I would like to thank you for your submission and the opportunity to analyze the paper. This article addresses a relevant and well-known topic in the field of sports physiotherapy. Although the authors present a "small" systematic review with few studies and low-quality evidence, it is important to note that it is well written and presents the reader with a critical summary of the current evidence on the topic.
However, I believe that some points need to be better presented and addressed in the text. I explain my concerns in more detail below. I ask that the authors specifically address each of my comments in their responses.
Answer: Dear reviewer, thank you for your valuable comments. We have tried to address them to the best of our abilities and believe that the manuscript has improved significantly on the basis of your suggestions, please see the manuscript attached.
- The introduction is well written, presents information about what you know about the topic, gaps and inconsistency. However, the authors create an expectation about HRV, which throughout the paper is frustrated
Response: We appreciate the time and effort that the Reviewer dedicated to providing feedback on our manuscript and are grateful for the insightful comments and valuable improvements to our paper. We agree with your comment. We designed our research strategy in an attempt to find studies that compare the effects of WBC and PBC on muscle performance, damage, soreness, and HRV. In fact, we did not find studies that matched our inclusion criteria evaluating the heart rate variability (HRV) outcome.
As we prospectively registered HRV as a secondary outcome in the International prospective register of systematic reviews - PROSPERO (registration number CRD42021246443), it is mandatory to conduct the review and report this absence of papers in this field. Finally, we have already discussed the HRV assessment and this limitation in the discussion section (Lines 432-438). Therefore, we would like to maintain this outcome (as a secondary) according to our prospective register of systematic reviews in PROSPERO.
- Do you consider it important to address this outcome in this review? During the planning of the review, did you not check if there were any more studies that had analyzed HRV?
Response: We consider it important to keep this outcome, because we performed a qualitative analysis of the included paper, and for the rationale of the review this outcome is an important pillar to explain physiological mechanisms involving WBC. We have discussed the parasympathetic modulation induced by WBC, according to the findings of Chauvineau et al. [18] (page 2 lines 71-73).
- Did the cited study (Chauvineau, 2021) not meet the eligibility criteria? This study is included in the review (Table 1).
Response: This study met the eligibility criteria and is presented in table 1 on page 6. In addition, this study was also included in the meta-analysis: Figures 2 (page 9), 4 (page 11), and 5 (page 12).
- The authors present the number of subjects evaluated. Then they present the information regarding the flowchart. Soon after, they present information about the subjects. I suggest presenting the flowchart data and then presenting the subjects' information
Response: We agree with the Reviewer's suggestion, and the number of subjects is presented in the flowchart data on page 5, figure 1.
- Correct the numbering of tables. You presented two tables 1.
Response: The table numbers have been corrected to table 1 on page 6 and table 2 on page 8.
- The databases presented in the flowchart differ from the databases presented in the method. Some bases are missing.
Response: We agree with the reviewer’s comment. Medline, Scielo, and Lilacs were not searched in this review. They were included in the methods section by mistake, and we apologize for this. The searched databases have been corrected on page 3, lines 116 – 117.
- Add information to table 1 (meaning of abbreviations).
Response: We agree with the reviewer’s comment. Abbreviations have been included in table 1 on page 6.
- I believe that more information about the studies is needed. Improve the description of studies. For DOMS and MVC, which muscles were analyzed? Quadriceps? Triceps surae? Both? others?
Response: For the MVC outcome, all the included papers analyzed knee extension torque and the results are reported on page 9, line 235-243. For DOMS the included papers did not provide this information about muscles or assessed segments. DOMS was assessed using a 10-centimeter analogue scale, as explained on line 271 through the general perception of the volunteer, regardless of body segment or muscles.
- Although the authors have described in the method that they perform analysis with random effects, it is possible to observe in Figures 2A and 5A an analysis with fixed effects.
Response: Thank you for your thoughtful comment. We have changed the analysis to random effects on page 9, figure 2A, and on page 12, figure 5A. In addition, the MD and CI values have been adjusted accordingly (lines 241 for 2A and 308 for 5A). Interestingly, the main results of the meta-analysis did not change, i.e., no significant difference was observed between WBC and PBC.
- Finally, the point that I consider most critical in the results session refers to the method used for meta-analysis. The authors present results using MD. However, I doubt whether all studies used the same scale to assess MVC. Some authors used the value in N, others in Nm, and others presented normalized values (%). In this case, it is correct to use the SMD method for meta-analyses. I suggest checking this point for all outcomes.
Response: We appreciate this suggestion. After double-checking all the outcomes, we changed the presentation of the results to SMD. Please find the changes in figure 2 for MVIC on page 9; figure 3 for VJP on page 10; the figure 4 for DOMS on page 11; and figure 5 for CK levels on page 12
- Regarding HRV, the information presented is incorrect. Although it is not possible to carry out a meta-analysis, I suggest that the authors present a qualitative description of the findings of Maxime Chauvineau's study.
Response: We are in agreement with your comment. Please find the results and discussion about HRV on lines 414-423.
- I believe that the discussion and conclusion are clearly presented. However, it is necessary to verify if with the correction of the analyzes it will be necessary to adjust some information.
Response: The changes in the meta-analysis did not alter the results, or conclusions.

Reviewer 2 Report
I enjoyed reading this systematic review and meta-analysis investigating the effectiveness of WBC and PBC on markers of functional, perceptual and physiological recovery following exercise. Please see my specific comments below.
Given the heterogeneity of the modalities included in the current investigation, I am unclear on the rationale for splitting treatments into WBC and PBC. Reference is made to differing underpinning mechanisms, and it is stated that this may explain some of the inconsistency within the literature, however the current format does not allow for this to be adequately explored (cold water immersion versus ice pack application versus whole body cryotherapy chamber for example). There also seems to be some inconsistency regarding how treatments are grouped (further detail below).
Line 39-40 It might be helpful to provide clarity in relation to the cryotherapy modalities here - terminology has been used interchangeably in some previous studies. For example, some studies use 'PBC' to refer only to cold chambers or cabins where the head (and sometimes hands) are not exposed to cold, but here it seems to refer also to CWI. Similarly, 'cold chamber' to many may imply WBC, so the distinction needs to be made clear.
Line 41 'different cryotherapy strategies could be superior to passive recovery' is a little vague. How were the strategies deemed to be superior in the previous review? You highlight the areas that were not assessed, but do not state how the findings were quantified.
Line 49-51 'The magnitude of these mechanisms depends on the intensity and time of the cold application and how it impacts the body.' Could the authors give some clarification here - it is important to know the direction of the relationship(s) mentioned.
Line 53-55 You mention the impact of hydrostatic pressure here in reference to WBC. In line with my first comment, WBC does not usually/always refer to an immersion protocol, and as such, there may not be a limited beneficial effect of hydrostatic pressure. Please ensure that you clarify terms early on.
Line 99-100 Excluding studies with an active or passive condition in addition to the cryotherapy modalities has likely meant a huge number of relevant studies have been excluded. If the aim of the review is to directly compare WBC and PBC, the inclusion of additional intervention groups would not impact the outcomes reported herein?
Line 101-102 Please explain the rationale for excluding studies utilising participants undertaking regular training/physical activity, when the target population identified at line 87 states 'healthy individuals and athletes'. If the exclusion refers to participants exercising during the study, then please make this distinction clear.
Line 194 It is stated that 1 paper was excluded based on the fact it did not compare LOCAL VS WBC - please can you clarify? Is local synonymous with PBC in this case?
Tables: There are 2 tables entitled 'Table 1', please amend. Similarly, please list studies in the same order in each for consistency.
Line 199 After stating that papers with additional interventions including 'passive comparators' would be excluded, it appears that the majority of included studies have a passive control/placebo group, yet other papers employing similar study designs have been excluded? There appears to be some inconsistency with decisions relating to inclusion/exclusion.
Line 199 Pournot et al 2010 (listed as 2011 in bibliography) does not appear to meet the inclusion criteria as it only utilises PBC, not WBC (although the protocol is listed under WBC in the table, the paper states immersion to iliac crest only)
Line 199 and 213 Cheauvineau needs to be amended to 'Chauvineau'
Line 313 States that there are similar effects, with no difference between modalities - clarification or explanation of these effects would give clear context for the reader here.
Line 320-321 It is stated that the findings have implications for sports medicine and rehabilitation, but this is not elaborated on. As it stands, this seems a vague statement.
Line 339-340 Does 'changed' imply a positive or negative effect?
Line 347 Which studies are you referring to here when you state 'In the same investigations'? This needs to be made a little clearer.
Line 378 The reference here should be written in text, rather than bracketed.
The conclusion section is quite vague - please state what the effect is/effects are, and how this information can/should inform practice moving forward.
Author Response
I enjoyed reading this systematic review and meta-analysis investigating the effectiveness of WBC and PBC on markers of functional, perceptual and physiological recovery following exercise. Please see my specific comments below. Given the heterogeneity of the modalities included in the current investigation, I am unclear on the rationale for splitting treatments into WBC and PBC. Reference is made to differing underpinning mechanisms, and it is stated that this may explain some of the inconsistency within the literature, however the current format does not allow for this to be adequately explored (cold water immersion versus ice pack application versus whole body cryotherapy chamber for example). There also seems to be some inconsistency regarding how treatments are grouped (further detail below)
Answer: Dear reviewer, thank you for the opportunity to resubmit the revised version of our manuscript. Your comments were highly thoughtful, allowing us to improve the quality of our manuscript. We have made changes according to your advice, please see the manuscript attached.
- Line 39-40 It might be helpful to provide clarity in relation to the cryotherapy modalities here - terminology has been used interchangeably in some previous studies. For example, some studies use 'PBC' to refer only to cold chambers or cabins where the head (and sometimes hands) are not exposed to cold, but here it seems to refer also to CWI. Similarly, 'cold chamber' to many may imply WBC, so the distinction needs to be made clear.
Response: We appreciate the time and effort that the Reviewer dedicated to providing feedback on our manuscript and are grateful for the insightful comments and valuable improvements to our paper. The division into two groups was made considering the methods section of each included paper. We considered PBC as the applications that do not involve the whole body (cold chambers and immersion until the iliac spine), and WBC was regarded as immersion above the iliac spine (immersion until the neck or immersion of the whole body, including the head). Thus, considering our rationale to split the WBC from the PBC, the cold chamber should be regarded as WBC. We have done our best to clearly describe this definition in the introduction section on page 1, lines 39-41.
- Line 41 'different cryotherapy strategies could be superior to passive recovery' is a little vague. How were the strategies deemed to be superior in the previous review? You highlight the areas that were not assessed, but do not state how the findings were quantified
Response: This is a fascinating comment. We considered it important to highlight these areas with regard to the findings of the studies separately in the literature. The systematic review conducted by Hohenauer et al. [6] shows that cryotherapy could improve DOMS more than passive recovery. We clarified these findings in the text on page 1, line 44. We should consider the limitation of this meta-analysis in view of the low methodological quality of included studies and the small number of included studies. Although this meta-analysis did not show a significant difference between WBC and PBC, there is a large lack of consensus in the literature about the outcomes analyzed in our review. Considering the findings in the literature and our meta-analysis, we can suggest that, for DOMS, any modality of cryotherapy is better than passive recovery. Future well-controlled RCTs are required to address the question concerning the best cryotherapy modality in terms of DOMS and functional improvements.
Line 49-51 'The magnitude of these mechanisms depends on the intensity and time of the cold application and how it impacts the body.' Could the authors give some clarification here - it is important to know the direction of the relationship(s) mentioned
Response: We agree with the Reviewer’s suggestion. This statement was made considering the systematic review results with meta-analysis published by Machado et al. [7] on Sports medicine. In this paper, the authors highlighted the dose-response relationship. We have tried to clarify this statement in lines 53-55.
- Line 53-55 You mention the impact of hydrostatic pressure here in reference to WBC. In line with my first comment, WBC does not usually/always refer to an immersion protocol, and as such, there may not be a limited beneficial effect of hydrostatic pressure. Please ensure that you clarify terms early on.
Response: We appreciate this insight. This has been highlighted in lines 39-41. In our consideration of WBC for this meta-analysis hydrostatic pressure is a relevant topic to mention. In this meta-analysis, we considered WBC as applications that immersed the participants above the iliac spine and considered PBC as applications that immersed or applied cryotherapy until the iliac spine. Therefore, when we talk about WBC, we are considering the immersion of the whole body, not necessarily including the head.
- Line 99-100 Excluding studies with an active or passive condition in addition to the cryotherapy modalities has likely meant a huge number of relevant studies have been excluded. If the aim of the review is to directly compare WBC and PBC, the inclusion of additional intervention groups would not impact the outcomes reported herein?
Response: We believe this will probably not impact the outcomes because regardless of applying other active interventions or control, the search in the databases did not return many papers comparing WBC and PBC. Moreover, we aimed to show the clinicians and physiologists more feasible ways to employ cryotherapy modalities, which are highlighted on page 2, lines 74 to 83.
- Line 101-102 Please explain the rationale for excluding studies utilising participants undertaking regular training/physical activity, when the target population identified at line 87 states 'healthy individuals and athletes'. If the exclusion refers to participants exercising duringthe study, then please make this distinction clear.
Response: We agree with the reviewer’s comments, and have changed the text in line 107 according to your request.
- Line 194 It is stated that 1 paper was excluded based on the fact it did not compare LOCAL VS WBC - please can you clarify? Is local synonymous with PBC in this case?
Response: We agree with the reviewer’s comment. We considered LOCAL and PBC as the same groups. The terminologies have been changed in the manuscript in figure 1 on page 5; and meta-analysis figures 2, 3, 4, and 5, on pages 9, 10, 11, and 12, respectively.
- Tables: There are 2 tables entitled 'Table 1', please amend. Similarly, please list studies in the same order in each for consistency.
Response: We agree with this observation, and the title of tables 1 and 2 have been corrected on pages 6 and 8, respectively.
- Line 199 After stating that papers with additional interventions including 'passive comparators' would be excluded, it appears that the majority of included studies have a passive control/placebo group, yet other papers employing similar study designs have been excluded? There appears to be some inconsistency with decisions relating to inclusion/exclusion.
Response: Dear reviewer, some included studies [5, 18, 12] compared intervention groups and passive or placebo groups. However, these studies were included because they also compared different cryotherapy modalities. A study was excluded when we were not able to find a comparison between interventions.
- Line 199 Pournot et al 2010 (listed as 2011 in bibliography) does not appear to meet the inclusion criteria as it only utilises PBC, not WBC (although the protocol is listed under WBC in the table, the paper states immersion to iliac crest only).
Response: We agree with this suggestion. This paper was included in table 1 by mistake, but you can observe that this paper was not included in the meta-analysis because the authors did not reply to our contact by email. We need the raw data from this study for it to be included in RevMan software to perform stats. You can find this report in figure 1 on page 5.
- Line 199 and 213 Cheauvineau needs to be amended to 'Chauvineau'
Response: The name has been amended in table 1 on page 6, and in table 2 on page 8.
- Line 313 States that there are similar effects, with no difference between modalities - clarification or explanation of these effects would give clear context for the reader here.
Response: We agree with this comment, and have clarified these effects by splitting them by outcomes in the text in lines 342-346.
- Line 320-321 It is stated that the findings have implications for sports medicine and rehabilitation, but this is not elaborated on. As it stands, this seems a vague statement
Response: We agree with this observation. After correcting the first paragraph of the discussion following your previous comment (lines 342-346 on page 12), these implications should be clearer to the reader.
- Line 339-340 Does 'changed' imply a positive or negative effect?
Response: We agree with this suggestion, the term “changed” has been replaced by “improved” on page 13, line 374.
- Line 347 Which studies are you referring to here when you state 'In the same investigations'? This needs to be made a little clearer.
Response: We agree with this comment. We have changed the sentence to make it clearer on page 13, line 381. In this sentence, we want to refer to the included studies.
- Line 378 The reference here should be written in text, rather than bracketed.
Response: We did not understand your request since the reference needs to be cited in brackets according to the Journal of Clinical Medicine (https://www.mdpi.com/journal/jcm/instructions). Please, let us know whether you need additional information or a request about this question.
- The conclusion section is quite vague - please state what the effect is/effects are, and how this information can/should inform practice moving forward.
Response: We agree with this comment. The conclusion has been adjusted to make it clearer concerning practice on lines 440-444.

Round 2
Reviewer 1 Report
I would like to congratulate the authors for the revised version. I believe that all points were duly addressed.
Author Response
Referee's comments:
Reviewer 1
- I would like to congratulate the authors for the revised version. I believe that all points were duly addressed.
Answer: Dear reviewer, thank you very much to recommend our manuscript to be accepted for publication in the Journal of Clinical Medicine.
Reviewer 2 Report
Thank you for addressing my initial comments, and for submitting a revised manuscript for review. Please see the points below for consideration;
1. Line 57-58. My previous comment in relation to hydrostatic pressure has not been fully addressed. Whilst I understand the differentiation you have now given in regards to WBC vs PBC, there are numerous studies using WBC in the form on chambers. These use cold air and are not immersion protocols, therefore there is no effect of hydrostatic pressure. Please clarify that you are referring to immersion protocols here, rather than cryo cabins or chambers, which would still be considered WBC using your definition.
2. Line 363 (previously line 378 - the reference here should be written in text, rather than bracketed). I am aware of the reference style of the journal and that references should be numbered in brackets, however, you need to include the author names in the text, as the sentence in it's current format is grammatically incorrect. 'Nonetheless, it is essential to highlight that PLEASE ADD NAME HERE [18] observed that WBC induced more significant...'
Author Response
Reviewer 2
- Line 57-58. My previous comment in relation to hydrostatic pressure has not been fully addressed. Whilst I understand the differentiation you have now given in regards to WBC vs PBC, there are numerous studies using WBC in the form on chambers. These use cold air and are not immersion protocols, therefore there is no effect of hydrostatic pressure. Please clarify that you are referring to immersion protocols here, rather than cryo cabins or chambers, which would still be considered WBC using your definition.
Answer: Dear reviewer, thank you for your valuable comments in this second round of the review process. We have tried to address them to the best of our abilities and believe that the manuscript has improved significantly on the basis of your suggestions.
We agree with your comment, we double checked the included papers and we observed that two of our included studies [12,13] used WBC in cold chamber modality. However, after a search in literature, we can infer that immersion or cold air applications, induced the same physiological effects through the induction of a noxious stimuli evoked by the cold [6, 14] . So we did our best to clarify the involvement of cold chambers in WBC in page 1, line 40; page 2 line 61-63; page 14 lines 392-398.
- Line 363 (previously line 378 - the reference here should be written in text, rather than bracketed). I am aware of the reference style of the journal and that references should be numbered in brackets, however, you need to include the author names in the text, as the sentence in it's current format is grammatically incorrect. 'Nonetheless, it is essential to highlight thatPLEASE ADD NAME HERE[18] observed that WBC induced more significant...'
Answer: Dear reviewer, we agree with this suggestion and the text was updated in page 14, now line 373 and the name was inserted in the sentence.
We believe that we have addressed all questions raised by the reviewers, and the manuscript has been greatly improved. Thank you for your time and effort in considering this manuscript for publication.